# Near-Hexaploid and Near-Tetraploid Aneuploid Progenies Derived from Backcrossing Tetraploid Parents *Hibiscus syriacus* × (*H. syriacus* × *H. paramutabilis*)

**DOI:** 10.3390/genes13061022

**Published:** 2022-06-06

**Authors:** Hsuan Chen, Ryan N. Contreras

**Affiliations:** 1Department of Horticultural Science, North Carolina State University, Raleigh, NC 27695, USA; 2Department of Horticultural, Oregon State University, Corvallis, OR 97331, USA; ryan.contreras@oregonstate.edu

**Keywords:** aneuploidy, infertility, interspecific hybridization, ornamental plant breeding, ploidy manipulation, polyploidy, unreduced gamete

## Abstract

*Hibiscus syriacus*, azalea, is an important woody ornamental shrub planted throughout many temperate and subtropical regions of the world. However, flower size is smaller in this species than some of its relatives. To increase flower size, interspecific hybridization has been used, and such hybrid cultivars are usually characterized by larger flowers, increased vigor, diverse leaf shapes, and reduced fertility. Our earlier studies have shown that these hybrid cultivars could backcross with *H. syriacus* when used as male parents. To understand the breeding potential of these hybrid cultivars, two popular tetraploid hybrid cultivars, ‘Lohengrin’ and ‘Resi’, were used as pollen parents to backcross several tetraploid *H. syriacus* cultivars. As a result, 28.76% and 64.4% of ‘Lohengrin’ and ‘Resi’ progenies exhibited larger flowers than both of their parents. Interestingly, 14 of 18 progenies of ‘Resi’ were putative hexaploids, whereas 19 tested ‘Lohengrin’ progenies were tetraploid. Because putative hexaploid progenies were only observed among progenies of ‘Resi’, this hybrid cultivar appears to produce unreduced gametes. In addition, among the 14 putative hexaploids derived from ‘Resi’, 11 had larger flowers than both of their parents and their tetraploid siblings (*p* < 0.05). The 45S rDNA and 5S rDNA locus segregation among those BC_1_F_1_ progenies was tested by fluorescent in situ hybridization (FISH), and the wide range of 45S rDNA signal numbers among siblings indicated that these aneuploids resulted from unequal segregation or chromosome rearrangement. Chromosome counting confirmed aneuploidy among BC_1_F_1_ progenies. Ploidy diversity and aneuploidy have been known to contribute to various elements of morphological diversity, such as larger flower size and reduced fertility, which are important in ornamental plant breeding. The present study demonstrated the breeding potential of interspecific *Hibiscus* cultivars for increasing ploidy level and flower size.

## 1. Introduction

*Hibiscus* species, within the Malvaceae family, have been ubiquitously used as ornamental plants due to their showy flowers, various growth habits, and long flowering season [1]. *Hibiscus* species can be categorized as tropical or hardy types, depending on their hardiness in temperate zones. Most *Hibiscus* species are tropical, but *H. syriacus* and *H. moscheutos* are widely found and used in temperate regions due to their broad adaptability. *Hibiscus syriacus* is the hardiest of the commonly used ornamental *Hibiscus* species and is used from the temperate zone to the subtropics due to its cold tolerance [2]. Not only is *H. syriacus* valued for its adaptability across USDA Hardiness Zones 5–9 [3], but it is also appreciated for its attractive garden performance and various bloom characteristics [2]. *H. syriacus* cultivars have a long blooming season from summer through fall, a wide range of flower types and colors, and tropical-looking flowers. For this species, high-impact flowers are the most important trait for the ornamental plant market; thus, breeders have largely focused on improving novel floral traits and flower size [4,5].

Interspecific hybridization has been a common strategy in *Hibiscus* to introgress novel traits and phenotypes and to exploit hybrid vigor [1,6,7,8,9,10,11]. Two related species, *H. sinosyriacus* and *H. paramutabilis*, have been used to improve *H. syriacus* due to their close genetic relationship and their cross-compatibility [6,7]. The resulting interspecific hybrids have larger flowers, varied leaf morphology, and increased vigor compared to their *H. syriacus* parents [6]. Two recently released hybrid cultivars, ‘Daewangchun’ and ‘Tohagol Red’, were made by conducting reciprocal crosses of *H. sinosyriacus* ‘Seobong’ and *H. syriacus* ‘Samchully’. ‘Daewangchun’ exhibits hybrid vigor, a uniquely extended red eyespot, and an excellent plant habit, whereas ‘Tohagol Red’ possesses a superb flower shape and color, with a distinct plant habit [10,11]. However, neither hybrid cultivar exhibited reduced fertility as expected. 

Interspecific hybridization between *H. paramutabilis* (2*n* = 4*x* = 82) and *H. syriacus* (2*n* = 4*x* = 80) represents another option that may be more feasible to increase flower size and reduce fertility [6]. *Hibiscus paramutabilis* is one of the only few woody ornamental *Hibiscus* species that thrives in temperate climates and possesses a larger flower than all of the 43 tested *H. syriacus* cultivars [7]. Interspecific hybridization between *H. paramutabilis* and *H. syriacus* is possible but has only been successful when *H. syriacus* is used as the female parent [7]. Resulting hybrid progeny had intermediate leaf morphology and larger flowers. Although the F_1_ hybrids had 65% to 88% apparently viable pollen, self-pollination led to few progeny [6]. Fortunately, a few F_2_ progenies were obtained, which indicated that these interspecific hybrids were not a dead-end breeding strategy [6,12]. Surprisingly, all of the F_2_ progenies were hexaploid (2*n* = 6*x* = 120), presumably from unreduced gametes [12], which suggests that polyploidization through hybridization is a viable means to increase flower size among hardy hibiscus. 

Some interspecific hybrid cultivars (*H. syriacus* × *H. paramutabilis*) including ‘Lohengrin’, ‘Resi’, and ‘Tosca’, have been in the market for decades, and all of them have large flowers and low or no fertility [3]. Our earlier study indicated that the hybrid cultivars have asymmetric infertility [4]. By using ‘Lohengrin’ and ‘Resi’ to reciprocally cross with *H. syriacus* cultivars, the results showed that the two hybrid cultivars are almost female-infertile, but their male fertility is not significantly different from fertile *H. syriacus* cultivars. When used as male parents in crosses with *H. syriacus* cultivars, ‘Lohengrin’ and ‘Resi’ produced 37% and 35% fruit per pollination, respectively, while, among fertile *H. syriacus* cultivars, the average was 43% [4]. The asymmetric infertility showed that the hybrid cultivars can be used in *H. syriacus* breeding; however, their value to increase flower size remains unknown.

Ploidy manipulation is an important breeding strategy for ornamental *Hibiscus* [13]. Changing ploidy could alter plant characters including sepals, petals, fruits, and seeds. In *Hibiscus*, ploidy manipulation has been used to alter habit and reduce the fertility of *H. acetosella* [14] and *H. moscheutos* [15]. Popular ornamental species have numerous ploidy levels, and there is evidence that the ornamental value of *Hibiscus* cultivars could be improved further by altering ploidy levels. *Hibiscus rosa-sinensis* is a popular tropical species with thousands of varieties [2]. Cultivars of *H. rosa-sinensis* have chromosome complements of 46, 54, 63, 68, 72, 77, 84, 90, 96, 112, 132, and 144, with the chromosome number positively correlated with flower size [16]. In addition, various ploidy levels of *H. syriacus* cultivars or breeding materials of tetraploid, pentaploid, and octoploid have been developed and showed altered morphologies [7,13,16]. For example, the hexaploid (2*n* = 120) cultivars have a larger flower [7] and decreased fertility [17] compared to tetraploid cultivars [18]. 

Cytogenetics tools, such as fluorescent in situ hybridization (FISH) and genome in situ hybridization (GISH), have been used to understand the genome structure of *H. syriacus* and to track chromosome segregation patterns following ploidy manipulation and interspecific hybridization [13,19]. Ribosomal DNA (rDNA) locus numbers detected by FISH have been used to understand genome evolution among species of several genera, including *Gossypium* spp. [20], *Lycoris* spp. [21], and *Cucumis* [22]. In *H. syriacus*, the number of rDNA loci has been used to assess ploidy level and as indication of aneuploidy [13]. Tetraploid, hexaploid, and octoploid *H. syriacus* cultivars had four, six, and eight 45S rDNA signals, respectively [13,19], whereas two, three, and four 5S rDNA signals were observed among these cytotypes [13]. Interestingly, putative pentaploid seedlings from interploidy hybrids between tetraploid and hexaploid have various rDNA signals ranging between the values of the two parents. The distribution of 45S rDNA numbers of putative pentaploids indicates that the tetraploid *H. syriacus* genome is an allotetraploid. The near-pentaploids were aneuploids due to the unequal chromosome segregation of their hexaploid parent, which behaved as a triploid during meiosis [13]. The 5S rDNA and 18S rDNA locus numbers of *H. paramutabilis* were four and ten, respectively [19]. GISH analysis has facilitated tracking the chromosome recombination of interspecific hybrid *Hibiscus* (*H. syriacus* × *H. paramutabilis*), and recombined chromosomes were observed in hexaploid F_2_ progeny [23]. However, replicable GISH signal detection for species with a high number of small chromosomes can be challenging. Tracking specific chromosomes or particular loci to observe segregation might provide clearer results. 

The objectives of this study were to (1) evaluate the breeding potential of interspecific hybrid cultivars to increase flower size, (2) measure the ploidy level of the BC_1_F_1_ progeny, and (3) determine if chromosome rearrangements occur in the hybrid genome by tracking rDNA loci.

## 2. Materials and Methods

### 2.1. Plant Material

Ten *Hibiscus* cultivars, including eight double flower *H. syriacus* cultivars and two interspecific hybrid cultivars of *H. syriacus* × *H. paramutabilis* (Table 1), were used for pollination tests and for creating a 294-plant BC_1_F_1_ population. Interspecific hybrid cultivars ‘Lohengrin’ and ‘Resi’ were used as male parents to backcross to *H. syriacus* cultivars to generate the BC_1_F_1_ population. Pollination number, fruit set number, progeny number, and crossing compatibilities of different crossing combinations were previously reported [4]. Hybrid plants were grown in 15 L containers filled with 100% unaged Douglas fir bark (Lane Forest Products, Eugene, OR, USA) amended with 20N–2.6P–10.0 K controlled release fertilizer (CRF) with micronutrients (Multicote 8, Haifa Chemicals, Ltd. Savannah, GA) incorporated at 7 kg·m^−3^, 5N–0.4P–3.3K fertilizer (Premix Plus, Nursery Connection. Hubbard, OR) incorporated at 10 kg·m^−3^, and Talstar nursery granular insecticide (FMC Corporation, Philadelphia, PA, USA) incorporated at 3 kg·m^−3^ at the Lewis Brown Farm in Corvallis, Oregon (USDA Zone 8b, 44.552979, −123.218964). 

Petal area and leaf morphology (Figure 1) were phenotyped during summer 2017. The original population was previously described [4]. Petal area, which is a proxy for flower size, was measured by using two first-day flowers of each plant, as previously described [4]. A total of 50 BC_1_F_1_ plants were selected in 2018 using quantile regression of flower size and petal number [4]. Randomly selected individuals from the 50 selected individuals were further used for flow cytometry analysis, chromosome counts, and FISH analysis. For the cytogenetics research, 18 ‘Resi’ and 19 ‘Lohengrin’ progenies were randomly chosen from a population previously selected for flower size; thus, the selected plants do not represent a truly random sample from all BC_1_F_1_ plants. 

### 2.2. Pollen Staining Test and Size Measurement

The pollen viability of ‘Lohengrin’, ‘Resi’, and *H. syriacus* ‘Red Heart’ was tested using 2% acetocarmine stain. *H. syriacus* ‘Red Heart’ (2*n* = 4*x*) represents a fertile allotetraploid *H. syriacus* for comparison with the two hybrid cultivars, ‘Lohengrin’ and ‘Resi’ (2*n* = 4*x*). Mature pollen was extracted from the flowers on the first day of their 2 day flowering period and then laid on a clean slide. One to two drops of 2% acetocarmine was mixed with pollen before the cover slide was applied. Pollen viability and pollen size were observed and measured under ×100 magnification using a compound light microscope (Axio Imager A1; Zeiss Microscopy, Oberkochen, Germany). Fully stained pollen was referred to as viable pollen, whereas partially stained pollen and empty pollen were defined as nonviable pollen. The pollen size distributions (mature pollen and immature pollen) of the two hybrid cultivars were expressed by histogram plots (Figure 2). Size comparisons of stainable pollen between the three cultivars were tested by one-way-ANOVA and Tukey’s HSD in R.

### 2.3. Developing and Confirming Hybrids

A total of 294 BC_1_F_1_ plants from 14 cross-combinations were obtained following controlled pollination of double flower *H. syriacus* cultivars with pollen from hybrid cultivars [4]. Numbers of plants from different crossing combination are listed in Appendix A. Hybridity was confirmed on the basis of male-parent-specific morphology (leaf shape, leaf width, and petiole length) and molecular markers. Primers of inter-simple sequence repeats (ISSRs) and sequence-related amplified polymorphism (SRAP) markers from other *Hibiscus* species [24,25,26] were used in this research (Appendix A). In the polymerase chain reaction (PCR), 55 cycles with Hot Start PCR-To-Gel Taq Mix, 2× (VWR, Solon, OH, USA) and 100 ng of genome DNA were combined for the final volume of 25 μL. A polymorphic marker for each set of parents was selected to screen the progenies of each crossing combination. Progeny presenting at least one male-parent-specific band in ISSR or SRAP analysis was referred to as a non-self-pollinated hybrid.

### 2.4. Flower Size

Petal area, which is a proxy for flower size, was measured using two first-day flowers of each plant. Cultivars and the 294 hybrid plants in the BC_1_F_1_ population were phenotyped from 6 July to 29 September 2017. The floral trait data were further used in selection for our breeding project. A total of 50 BC_1_F_1_ plants were selected in 2018 using quantile regression of flower size and petal number [4]. Randomly selected individuals from the 50 selected individuals were further used for flow cytometry analysis, chromosome counts, and FISH analysis. The petal area of ‘Resi’ progenies with different putative ploidy levels (See below) were then compared by Wilcoxon signed-rank test.

### 2.5. Flow Cytometry

DNA content of the two hybrid cultivars and randomly chosen 18 ‘Resi’ progenies and 19 ‘Lohengrin’ progenies from the plants selected in the spring of 2018 were tested by flow cytometry (Table 1). For each sample, three young leaves were collected. *Solanum lycopersicum* ‘Stupicke’ (2C = 1.96 pg) was used as an internal standard to calculate nuclear DNA content [27,28]. For each sample, a 0.5 cm^2^ leaf segment of the hybrid plant and internal standard plant were co-chopped using a new razor with 400 μL of nuclei extraction buffer (Cystain Ultraviolet Precise P Nuclei Extraction Buffer; Sysmex, Görlitz, Germany). The chopped sample was filtered using a 30 μm gauge filter (Partec Celltrics, Münster, Germany) and collected in a 3.5 mL plastic tube (Sarstedt Ag & Co.; Nümbrecht, Germany); then, 1600 μL of stain buffer (Cystain Ultraviolet Precise P Stating Buffer; Sysmex, Görlitz, Germany) was added to the tube. The nuclei were then analyzed using a flow cytometer (Partec Ploidy Analyser PA-II; Partec).

The putative ploidy of each plant was defined by the genome size estimated by flow cytometry compared to a known tetraploid. Any individual with a genome size within 10% of that of the tetraploid parent was assigned as a putative tetraploid. Similarly, any individual with a genome size within 10% of the estimated hexaploid size was assigned as a putative hexaploid.

### 2.6. Chromosome Preparations

Chromosome squashes were performed on the two hybrid cultivars and 15 BC_1_F_1_ hybrid progenies. Fresh root tips were collected around noon time and emerged in the solution with 2 mM 8-hydroxyquinoline and 0.24 mM cycloheximide at 4 °C for 3 h, followed by transfer into Farmer’s solution (3 ethanol:1 acetic acid, by volume), in a 25 mL glass bottle overnight at 25 °C. The following day, root tips were transferred into 70% ethanol for long-term storage at 4 °C in a refrigerator. Enzyme digestion was performed on the slide to prevent loss of material during pipette transfer [29]. Enzyme-digested root tips were then squashed by the modified chromosome preparation method from Chang’s protocol [30]. Two drops of modified Farmer’s solution (3 methanol:1 acetic acid, by volume) was applied on the center of each slide, and root tip cells were dispersed by lightly tapping with a metal spatula. Four drops of modified Farmer’s solution was added to each corner of the slide before igniting the slide by running it through an alcohol lamp. Slides were allowed to dry in a 37 °C oven overnight. Dried slides were treated for 15 min in a diluted Giemsa stain (Sigma-Aldrich, Saint Louis, Montana, USA), rinsed in water, and allowed to dry at 37 °C in an oven. Stained slides were screened for condensed chromosomes at a magnification of ×1000 on a compound light microscope (Axio Imager A1; Zeiss). 

### 2.7. FISH Analysis

High-quality slides with multiple metaphase cells were selected for FISH analysis. Giemsa stain was removed by incubating slides in −20 °C Farmer’s solution for 5 min, followed by 5 min incubations in 75%, 95%, and 100% ethanol. Slides were then air-dried and stored at room temperature for up to 2 weeks. Synthesis of probes for FISH was carried out according to Chang’s protocol [30]. A plasmid DNA construct from wheat, pTA794, containing the 5S rDNA repeat [31], was labeled with digoxigenin (DIG-11-dUTP) using DIG-Nick Translation Mix (Roche Diagnostics; Mannheim, Germany). The digoxigenated probe was represented by a red fluorescent signal, detected using Texas red (anti-digoxigenin-rhodamine Fab fragments) (14877500; Roche Diagnostics). Another plasmid DNA from wheat, pTA71, containing ~9 kb of coding sequences from the 45S (18S-5.8S-26S) rDNA gene [32], was labeled with biotin (Biotin-16-dUTP) by Biotin-Nick Translation Mix (Roche Diagnostics). The biotinylated probe was represented by a green fluorescent signal, detected using fluorescein anti-biotin (SP-3040; Vector Laboratories, Burlingame, CA, USA). Counterstaining of chromosomes was performed with 4′,6-diamidino-2-phenylindole (DAPI) suspended in a mounting medium at 1.5 μg·mL^−1^ (Vectashield; Vector Laboratories, Burlingame, CA, USA).

Chromosome labeling and probe detection for FISH analysis were carried out according to previous methods [33], and pepsin treatment was modified for specific plant materials in this study. Briefly, 0.1% pepsin in 10 mM HCl (*w*/*v*) was applied to each slide (150 μL per slide) and covered with a plastic coverslip. Pepsin treatments were carried out at 37 °C for 1–1.5 h. 

## 3. Results

### 3.1. Pollen Size Distribution

The mean sizes of viable pollen of *H. syriacus* ‘Red Heart’, ‘Lohengrin’, and ‘Resi’ were 155.40 ± 0.45 µm, 148.74 ± 1.78 µm, and 150.11 ± 0.54 µm (mean ± 95% confidence interval). No size difference between the two hybrid cultivars was found (*p* > 0.05); however, the pollen size of ‘Red Heart’ was significantly larger than that of ‘Lohengrin’ and ‘Resi’. Moreover, extra-large pollen (>180 µm) was observed in ‘Resi’ (6%) but rarely observed in ‘Lohengrin’ and *H. syriacus* ‘Red Heart’ (Figure 2).

### 3.2. Hybrid Confirmed by Leaf Morphology and ISSR

Leaf morphology and molecular markers confirmed that all BC_1_F_1_ seedlings were the result of hybridization. Leaf morphology between *H. syriacus* and the two hybrid cultivars was clearly distinguishable (Figure 1). Petiole lengths of *H. syriacus* cultivars were short (1–2 cm), while petiole lengths of the hybrid cultivars, ‘Lohengrin’ and ‘Resi’, were longer (3–8 cm). Like hybrid cultivar parents, all of the BC_1_F_1_ progenies had a longer petiole and larger leaves (Figure 1). In addition, all progenies in the study showed at least one male-parent-specific PRC fragment in the ISSR or SRAP test (Appendix A).

### 3.3. Flow Cytometry Analysis for Ploidy

Putative tetraploid and hexaploid progenies were identified, even though all the parents were tetraploid. ‘Resi’ yielded 78% near-hexaploid progeny, whereas all progenies of ‘Lohengrin’ were near-tetraploid (Table 2). Genome size variation, confident intervals, and results of Tukey’s honestly significant difference (HSD) test (α = 0.05) for genome size comparisons are listed in Appendix A. Genome sizes of the two hybrid cultivars were very close to standard tetraploid *H. syriacus* cultivars. The 19 BC_1_F_1_ individuals from ‘Lohengrin’ were all putative tetraploid (Table 1). Even though our results indicated a high percentage of randomly picked progenies from ‘Resi’ as hexaploid, these plants were obtained through an early selection based on vigor and flower size. Thus, the observation of 78% progenies being hexaploid may not be representative of a random unselected population.

### 3.4. Flower Size Distribution and Other Phenotypes

In the hybrid population of 294 BC_1_F_1_ plants, unexpectedly high percentages of the hybrid cultivar progenies had a larger petal area than their parents (Figure 3). Among the 233 ‘Lohengrin’ progenies, 67 (29%) progenies had a larger petal area than ‘Lohengrin’. A greater percentage, 38 of 59 (64%), of ‘Resi’ progenies had a greater petal area than ‘Resi’. The results revealed that the hybrid cultivars could be great breeding materials for petal size enlargement.

The petal size difference between putative tetraploid and putative hexaploid progenies of ‘Resi’ was further analyzed. Among the 14 putative hexaploid progenies derived from ‘Resi’, 11 (79%) had a larger petal area than ‘Resi’, while only two (50%) of the four confirmed tetraploid progenies had a larger petal area than ‘Resi’. The average petal area size of the 14 putative hexaploid ‘Resi’ progenies was 43.0 cm^2^, while that of the four putative tetraploid was 35.2 cm^2^. According to Wilcoxon’s signed-rank test, the petal area of putative hexaploids was significantly larger than that of putative tetraploids (*p* = 0.0498). According to Wilcoxon’s signed-rank test upon removing the outliner H2015-024-08, the petal area size of putative hexaploids was significantly larger than that of tetraploids (*p* = 0.024). Considering that the nonrandom and small size of the population, a larger-scale test might be needed. The result preliminarily showed that a higher ploidy level generally increased petal area (Figure 3), but a larger-scale test in a random population would be needed to confirm the relationship. Nevertheless, ‘Resi’ yielded a significant number of hexaploid progenies as a pollen parent, demonstrating its value for ploidy manipulation facilitated by unreduced gametes. 

### 3.5. Chromosome Squash and FISH Analysis

Chromosome counts indicated that BC_1_F_1_ progenies were aneuploid with chromosome numbers ranging from 122 to 130. For example, H2015-016-3 was a near-hexaploid accession with 128 chromosomes (Figure 4), whereas a true hexaploid would be expected to have 120 or 121 chromosomes. The FISH analysis further confirmed aneuploidy of hybrid progenies on the basis of the 45S rDNA number distribution. The 45S and 5S rDNA locus numbers varied among the two hybrid cultivars and 15 selected BC_1_F_1_ progenies (Table 2). The 45S and 5S rDNA locus numbers of tetraploid *H. syriacus* cultivars were four and two. The 45S and 5S rDNA locus numbers of the two hybrid cultivars were all five and two. A wider than expected range of 45S rDNA locus numbers was observed among the hybrid progenies, but the number of 5S rDNA loci followed their putative ploidies (Table 2). Among tetraploid progenies, the locus numbers of 45S ranged from four to six, while the locus number of 5S was two (Figure 5 and Table 2). Among hexaploid progenies, the locus numbers of 45S ranged from five to eight, while the locus number of 5S was three (Figure 6 and Table 2).

## 4. Discussion

### 4.1. Pollen Size Distribution and Source of Unreduced Gametes

The hybrid cultivar ‘Resi’ was shown to be a valuable male parent for *Hibiscus* ploidy manipulation. An unexpected number of ‘Resi’ seedlings were hexaploid; however, all parent taxa were previously confirmed as tetraploids. The putative hexaploid progeny resulted from unreduced gametes from ‘Resi’, which produced a low but detectable percentage of unreduced pollen. Crossing with the same group of female parents, 78% of tested ‘Resi’ seedlings were putative hexaploid; however, no putative hexaploid offspring of ‘Lohengrin’ were found (Table 2). If the unreduced gamete was from the female parents, a certain percentage of hexaploid ‘Lohengrin’ would have been found. The pollen size analysis confirmed the source of unreduced gametes. Although the average sizes of stainable pollen of the tested tetraploid *H. syriacus* ‘Red Heart’ and the two hybrid cultivars were all very similar, some extra-large pollen was only observed from ‘Resi’. About 6% of ‘Resi’ pollen was extra-large, which was very rarely observed in ‘Lohengrin’ and *H. syriacus* ‘Red Heart’ (Figure 2). The evidence of hexaploid seedlings combined with extra-large pollen from ‘Resi’ confirmed it as the source of unreduced gametes. Data showed that ‘Resi’ is a good parent for ploidy manipulation via hybridization; however, our earlier research showed that it has a lower proportion of stainable pollen than other cultivars and has a slightly lower expected number of seedlings received per pollination [4]. The exact percentage of hexaploid seedlings might be higher or lower than the observed 78% because the population was first selected for petal area during 2016. In *H. rosa-sinensis*, a positive relationship between genome size and flower size was found [16]. Thus, we recommend ‘Resi’ as a pollen parent to breed for larger flowers, not only through transgressive segregation but also ploidy manipulation. Breeders wishing to increase flower size using ploidy manipulation can use treatment of somatic material in vitro or in vivo to create octoploids and backcross to yield hexaploids [18]. However, using ‘Resi’ as a male parent and selecting progeny with large flowers may be a more efficient method.

Different interspecific hybrids of *H. syriacus* × *H. paramutabilis* might have different potential in *Hibiscus* cultivar improvement due to the varied fertility and frequency of unreduced gametes. This study indicated that the hybrid cultivars ‘Resi’ and ‘Lohengrin’ clearly varied in both these aspects, with ‘Resi’ having lower male fertility but higher frequency of unreduced gametes. In earlier research, an unreduced egg cell was expected to be the source of unreduced gametes [12]. Low fertility of F_1_ hybrids of *H. syriacus* × *H. paramutabilis* has been reported, which yielded only six F_2_ plants from a few thousand pollinations. Interestingly, all F_2_ seedlings were hexaploid. Because no unreduced pollen was found using flow cytometry analysis, unreduced eggs were identified to be the cause of the hexaploid progeny [12]. Although the interspecific hybrids’ value for ploidy manipulation was confirmed, the low female fertility of the hybrid limits its applicability [12]. However, in the present study, ‘Resi’ was observed to produce unreduced male gametes and yielded only hexaploid progenies. 

Unreduced gametes have been used for ploidy manipulation of clonal crops such as cassava (*Manihot esculenta*) [34] and lily (*Lilium* sp.) [21]. In addition to ploidy manipulation to increase flower size, unreduced gametes might help to circumvent hybrid infertility, which can be an important consideration when making wide hybrids. Interspecific or heteroploidy hybrids are often infertile, but they might occasionally produce functional unreduced gametes that could be used for further hybridizations [35,36,37,38]. For example, in an F_1_ interspecific hybrid of *Lilium auratum* (2*x*) *× L. henryi* (2*x*), a 2% acetocarmine stain test showed that 42% of stainable pollen was unreduced gametes. Furthermore, all BC_1_F_1_ seedlings were triploid, which implies that only unreduced gametes were fertile [30,39]. To conclude, unreduced pollen from interspecific hybrids avoids time-consuming ploidy manipulation treatments to efficiently provide increase ploidy level, but may also facilitate gene introgression when the fertility of the F_1_ is otherwise low. 

### 4.2. Breeding Potential of Hybrid Cultivars to Increase Flower Size

The petal area distribution of BC_1_F_1_ seedlings illustrates the utility of hybrid cultivars to increase flower size (Figure 3). This supports previous reports that recommended interspecific hybridization of *H. syriacus × H. paramutabilis* to obtain large flowering hybrids [6]. Flower sizes of these F_1_ hybrids were larger than those of both parents [6], and our study found that hybrid cultivars have the potential to further increase flower size via transgressive segregation and ploidy manipulation. Nearly 29% and more than 64% of seedlings of ‘Lohengrin’ and ‘Resi’, respectively, had larger flowers than their parents. Transgressive segregation likely accounts for some of this increase in flower size. However, a surprisingly high percentage of ‘Resi’ progenies had larger flowers than both of their parents. Ploidy level contributed to increased flower size, as demonstrated by near-hexaploids exhibiting a larger petal area than near-tetraploids among the ‘Resi’ progeny. Thus, both transgressive segregation and ploidy level contributed to increased flower size. 

### 4.3. Diversity of rDNA Signals among BC_1_F_1_ Seedlings

The diversity of the 45S rDNA signal numbers indicated that meiotic irregularities such as improper pairing and/or segregation occurred, which may provide further phenotypic diversity. The 45S rDNA number of *H. syriacus* corresponds to their ploidy such that there are four 45S rDNA loci in tetraploids and six 45S rDNA loci on hexaploids [13,19]. In our study, the 45S rDNA signal number of the two hybrid cultivars was five. Without abnormal segregation, for tetraploid progenies, the expected signal number of 45S should be between the parents. In the present study, an unexpectedly wider range of 45S rDNA signal numbers was found. Among tetraploid seedlings, four to six 45S rDNA signals were detected, whereas, among hexaploid seedlings, five to eight signals were observed. This result indicates that some abnormal segregation events occurred during meiosis of the two hybrid cultivars. The first possible explanation of the variation could be that the 45S rDNA locus is located on a different chromosome in the genomes of *H. syriacus* and *H. paramutabilis*. A genome with the 45S rDNA loci on an unmatched chromosome would produce gametes with various numbers of 45S rDNA. Another possible explanation would be abnormal chromosome pairing or chromosomes with 45S rDNA loci from two species simply not pairing well in meiosis, which would result in random segregation. Interestingly, the signal numbers of 5S rDNA always correspond to ploidy, with two signals for near-tetraploids and three signals for near-hexaploids. This implies that chromosomes with 5S rDNA loci from *H. syriacus* and *H. paramutabilis* genomes are homologous, which allows normal pairing and segregation in the hybrid genome. 

The genome size and chromosome-count data also confirmed the possibility of abnormal chromosome segregation. Every BC_1_F_1_ seedling had a genome size close to a euploid tetraploid or hexaploid genome size; however, the genome sizes of all progenies were slightly larger than those of the *H. syriacus* standard and the two hybrid cultivars (Table 2 and Appendix A). Theoretically, tetraploid progenies should have a genome size approximately equal to the average of their two parents, whereas hexaploid progenies should have a genome size of approximately 1.5 times the tetraploid genome size. However, the genome sizes of 23 near-tetraploids and 14 near-hexaploids were all close but greater than the average genome sizes of their parents. The result implies that gametes with a bigger genome size might be more competitive during pollination or have a greater chance of survival. Chromosome counting confirmed the genome size results. The chromosome numbers of *H. syriacus* and *H. paramutabilis* genomes were reported to be 80 and 82 [6]. Thus, theoretically, hybrid BC_1_F_1_ seedlings should be tetraploid with 80–81 chromosomes or be hexaploid with 120–122 chromosomes if all chromosomes were paired well. However, BC_1_F_1_ seedlings in the present study had 80–88 chromosomes (tetraploid or near-tetraploid aneuploids) or 120–130 chromosomes (hexaploidy or near-hexaploid aneuploids; Figure 4). Larger genome sizes resulting from extra chromosomes among the BC_1_F_1_ seedlings were observed, and abnormal segregation and diversity in chromosome number were also confirmed.

Combining the results of rDNA signal distribution and chromosome counting, we suggest that the meiosis of hybrid cultivars exhibits chromosome pairing errors or chromosome rearrangement. Traditionally, incomplete chromosome sets (aneuploidy) could lead to low fertility and abnormal phenotypes, which might have negative effects on seed crops [40]. However, in clonally propagated ornamental plants, low fertility and novel phenotypes are often preferred; thus, creating aneuploid cultivars could be a good strategy. Actually, aneuploid manipulation has been used in clonal plants, including *Rubus* [41], *Saccharum* sp. (sugarcane) [42], and *Lilium* sp. (lily) [43]. Since tetraploid *H. syriacus* has been reported to have a disomic segregating allotetraploid genome, the segregation of hexaploids would act as triploids, which might be linked with low fertility [13,44]. We report that the population developed from hybrid cultivars produce ploidy and chromosomal level diversity that might potentially provide novel phenotypes. Our crossing strategy led to progenies with diversity in ploidy levels including aneuploids, which could be valuable for clonal and ornamental plant breeding. Our results also indicated that interspecific hybrid *Hibiscus* cultivars are useful breeding materials to increase flower size through transgressive segregation and ploidy manipulation, despite their relatively low male fertility.

## Figures and Tables

**Figure 1 genes-13-01022-f001:**
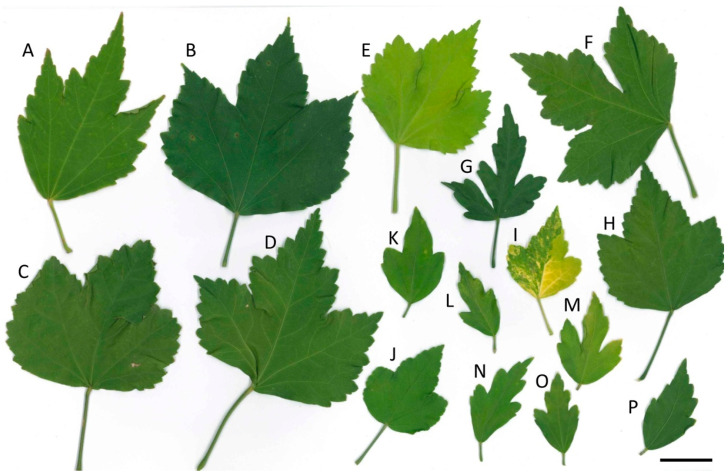
Leaf morphology of the *Hibiscus* hybrids and *H. syriacus*. (**A**,**B**) Hybrid cultivar pollen parents (**A**) ‘Lohengrin’ and (**B**) ‘Resi’. (**C**–**F**) Hexaploid BC_1_F_1_ hybrids (**C**) H2015-104-05, (**D**) H2015-024-08, (**E**) H2015-017-05, and (**F**) H2015-024-07. (**G**–**J**) Tetraploid BC_1_F_1_ hybrids (**G**) H2015-052-X2, (**H**) H2015-052-02, (**I**) H2015-064-10, and (**J**) H2015-062-08. (**K**) Hexaploid *H. syriacus* cultivar ‘Flogi’. (**L**–**P**) Tetraploid *H. syriacus* (**L**) ‘Blushing Bride’, (**M**) ‘Lavender Chiffon’, (**N**) ‘Blue Chiffon’, (**O**) ‘White Chiffon’, and (**P**) ‘Red Heart’. Scale bar = 3 cm.

**Figure 2 genes-13-01022-f002:**
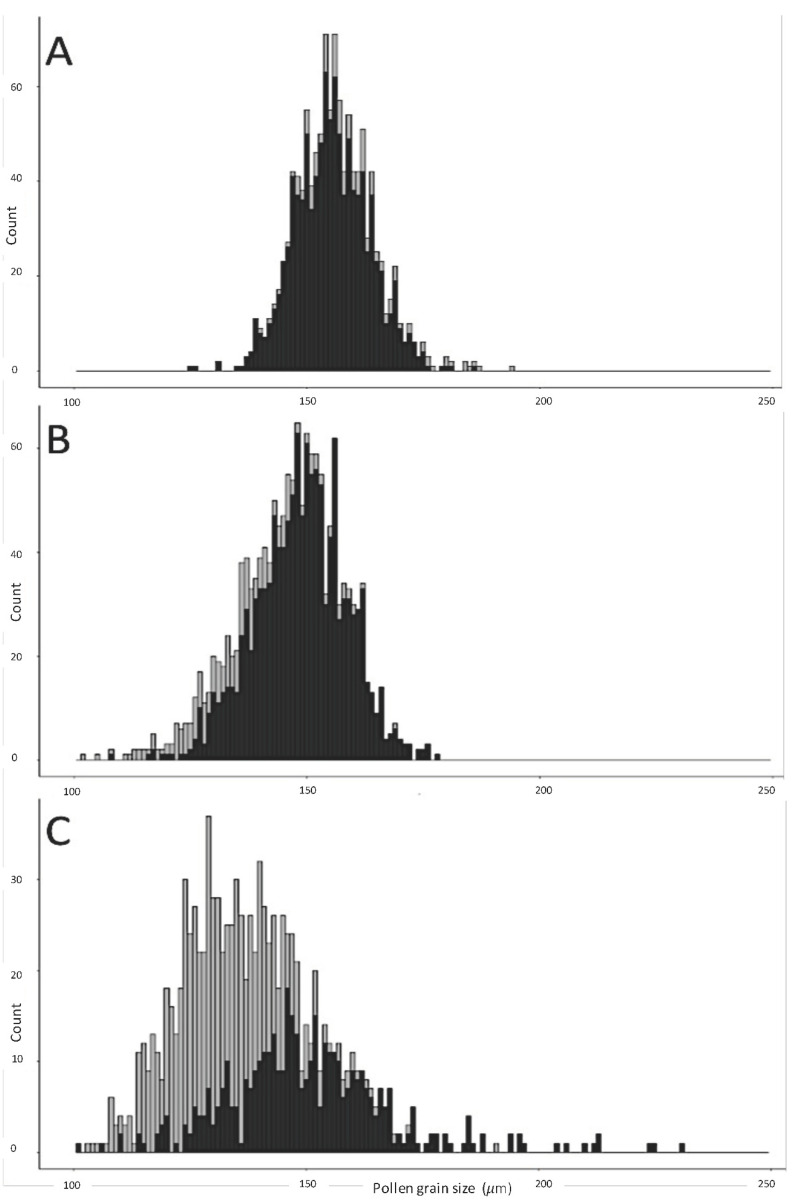
Pollen grain size (µm) distributions: (**A**) *H. syriacus* ‘Red Heart’, (**B**) *Hibiscus* ‘Lohengrin’, and (**C**) *Hibiscus* ‘Resi’. The black histogram indicates the counts of stainable pollen grains tested by 2% acetocarmine stain, while the gray histogram indicates empty pollen grains.

**Figure 3 genes-13-01022-f003:**
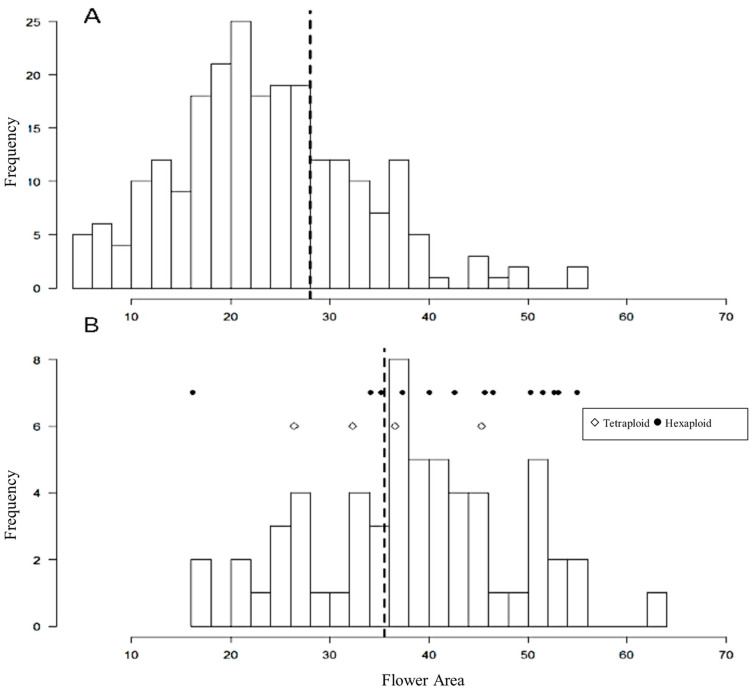
Petal area (cm^2^) distribution of BC_1_F_1_ seedlings (*H. syriacus* × (*H. syriacus* × *H. paramutabilis*)) of (**A**) ‘Lohengrin’ and (**B**) ‘Resi’. The dashed lines indicate the petal areas of ‘Lohengrin’ and ‘Resi’ in (**A**,**B**), respectively. Petal areas of each known ploidy ‘Resi’ progeny are labeled in the (**B**) figure. Flower area sizes and ploidies of plants listed in Table 1 are independently represented. The filled circles represent putative hexaploids, while open diamonds represent putative tetraploids. Each icon independently represents one individual listed in Table 1. Filled circles represent putative hexaploids, while open diamonds represent putative tetraploids. According to Wilcoxon’s signed-rank test upon removing the outliner H2015-024-08 (the leftmost one), the average petal area size of putative hexaploids was significantly larger than that of tetraploids (*p* = 0.024).

**Figure 4 genes-13-01022-f004:**
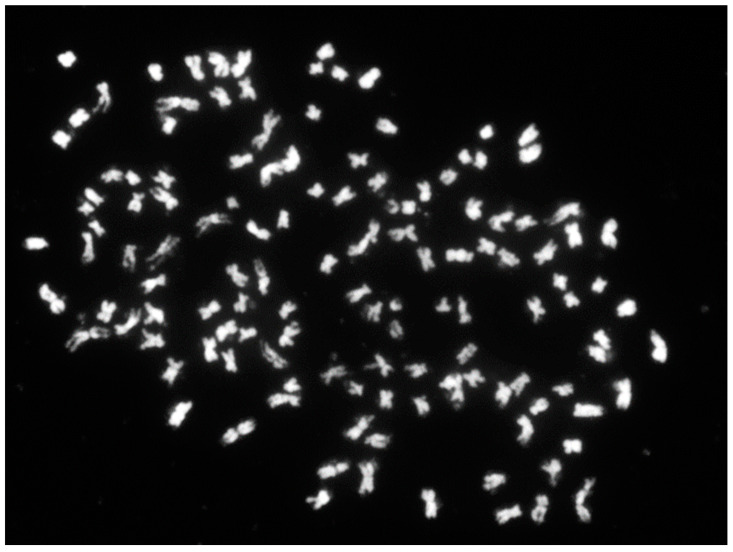
Chromosome squash of a BC_1_F_1_ interspecific hybrid Hibiscu*s* (*H. syriacus* (2*n* = 4*x* = 80) × (*H. syriacus* (2*n* = 4*x* = 80) × *H. paramutabilis* (2*n* = 4*x* = 82)), H2015-016-03 (2*n* = 6*x* + 8 = 128), with 128 metaphase chromosomes. Chromosomes were counterstained with DAPI. Scale bar = 50 μm.

**Figure 5 genes-13-01022-f005:**
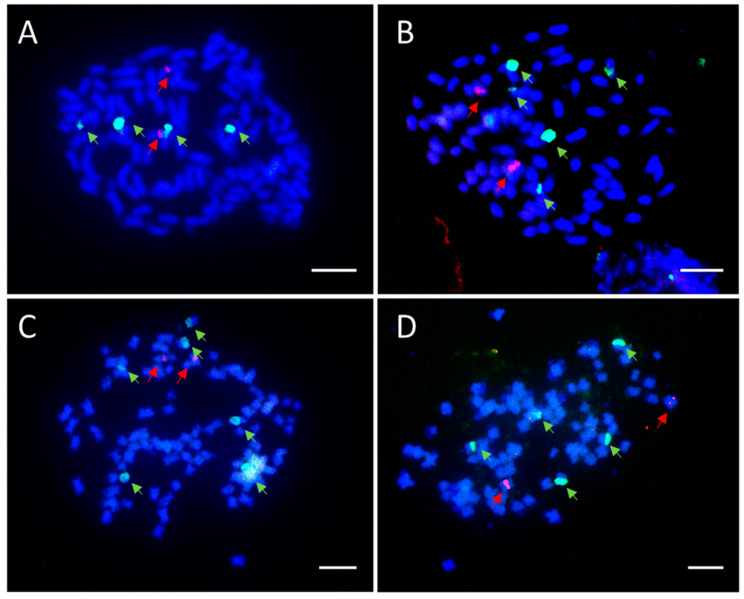
Results of FISH analysis of rDNA signals in mitotic cells of tetraploid (4*x*) BC_1_F_1_ seedlings (*H. syriacus* × *H.* ‘Lohengrin’ (*H. syriacus* × *H. paramutabilis*)). The cells were dual-probed with biotin-labeled 45S rDNA (green) and dig-labeled 5S rDNA (red) from wheat (*T. aestivum*) [33]. Chromosomes were counterstained with DAPI (blue). (**A**) H2015-029-02 with four 45S rDNA and two 5S rDNA signals. (**B**) H2015-061-03 with five 45S rDNA and two 5S rDNA signals. (**C**) H2015-050-X2 with six 45S rDNA and two 5S rDNA signals. (**D**) ‘Lohengrin’ with five 45S rDNA and two 5S rDNA signals. Scale bar in (**A**–**D**) = 50 μm.

**Figure 6 genes-13-01022-f006:**
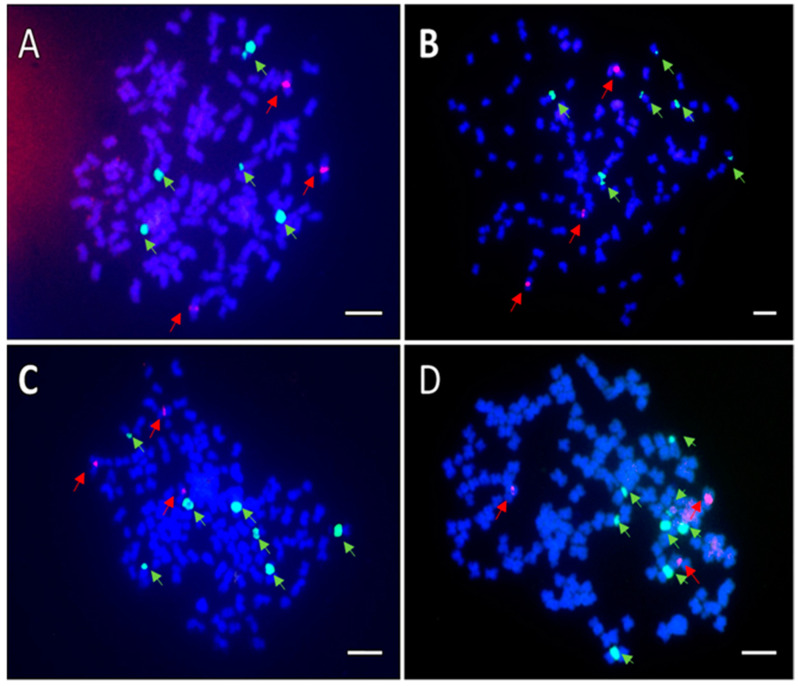
Results of FISH to analysis of rDNA signals in mitotic cells of hexaploid (6*x*) BC_1_F_1_ seedlings (*H. syriacus* × (*H. syriacus* × *H. paramutabilis*)). The cells were dual-probed with biotin-labeled 45S rDNA (green) and dig-labeled 5S rDNA (red) from wheat (*T. aestivum*). Chromosomes were counterstained with DAPI (blue). (**A**) H2015-016-08 with five 45S rDNA and three 5S rDNA signals. (**B**) H2015-109-01 with six 45S rDNA and three 5S rDNA signals. (**C**) H2015-016-01 with seven 45S rDNA and three 5S rDNA signals. (**D**) H2015-104-5 with eight 45S rDNA and three 5S rDNA signals. Scale bar in (**A**–**D**) = 50 μm.

**Table 1 genes-13-01022-t001:** Genome size and ploidy level of *H. syriacus* cultivar standard, interspecifichybrid cultivars, ‘Lohengrin’ and ‘Resi’, and BC1F1 hybrids ^1^.

Cultivar or Accession	Female Parent	Male Parent	Predicted Ploidy Level	Relative 2C Genome Size, Mean ± SE (pg)	Petal Area ^2^ (cm^2^)
*H. syriacus* ‘Fiji’			4*x*	4.63 ± 0.05	-
‘Resi’			4*x*	4.63 ± 0.08	35.5
‘Lohengrin’			4*x*	4.58 ± 0.03	28.0
H2015-019-08	*H. syriacus* ‘Blue Chiffon’	‘Resi’	6*x*	7.05 ± 0.01	53.0 *
H2015-019-09	*H. syriacus* ‘Blue Chiffon’	‘Resi’	6*x*	6.96 ± 0.05	54.9 *
H2015-024-01	*H. syriacus* ‘Blue Chiffon’	‘Resi’	6*x*	6.98 ± 0.02	40.0 *
H2015-024-05	*H. syriacus* ‘Blue Chiffon’	‘Resi’	6*x*	7.02 ± 0.05	34.1
H2015-024-07	*H. syriacus* ‘Blue Chiffon’	‘Resi’	6*x*	6.97 ± 0.03	45.7 *
H2015-024-08	*H. syriacus* ‘Blue Chiffon’	‘Resi’	6*x*	6.99 ± 0.03	16.2
H2015-016-01	*H. syriacus* ‘Blushing Bride’	‘Resi’	6*x*	7.05 ± 0.03	51.5 *
H2015-016-11	*H. syriacus* ‘Blushing Bride’	‘Resi’	6*x*	7.21 ± 0.09	42.6 *
H2015-016-02	*H. syriacus* ‘Blushing Bride’	‘Resi’	6*x*	7.07 ± 0.03	50.2 *
H2015-016-03	*H. syriacus* ‘Blushing Bride’	‘Resi’	6*x*	7.16 ± 0.08	35.2
H2015-016-05	*H. syriacus* ‘Blushing Bride’	‘Resi’	6*x*	7.04 ± 0.08	46.5 *
H2015-016-08	*H. syriacus* ‘Blushing Bride’	‘Resi’	6*x*	7.03 ± 0.00	37.3 *
H2015-017-05	*H. syriacus* ‘Blushing Bride’	‘Resi’	6*x*	6.94 ± 0.07	42.6 *
H2015-109-01	*H. syriacus* ‘Lavender Chiffon’	‘Resi’	6*x*	7.09 ± 0.08	52.6 *
H2015-108-02	*H. syriacus* ‘Lavender Chiffon’	‘Resi’	4*x*	4.76 ± 0.07	36.6 *
H2015-122-01	*H. syriacus* ‘Lavender Chiffon’	‘Resi’	4*x*	4.82 ± 0.03	45.3 *
H2015-122-02	*H. syriacus* ‘Lavender Chiffon’	‘Resi’	4*x*	4.83 ± 0.07	32.3
H2015-122-03	*H. syriacus* ‘Lavender Chiffon’	‘Resi’	4*x*	4.87 ± 0.05	26.4
H2015-029-02	*H. syriacus* ‘Strawberry Smoothie’	‘Lohengrin’	4*x*	4.72 ± 0.03	25.9
H2015-031-04	*H. syriacus* ‘Blushing Bride’	‘Lohengrin’	4*x*	4.72 ± 0.00	15.3
H2015-043-06	*H. syriacus* ‘Blushing Bride’	‘Lohengrin’	4*x*	4.75 ± 0.02	19.7
H2015-043-07	*H. syriacus* ‘Blushing Bride’	‘Lohengrin’	4*x*	4.83 ± 0.08	22.9
H2015-044-17	*H. syriacus* ‘Blushing Bride’	‘Lohengrin’	4*x*	4.73 ± 0.07	18.7
H2015-046-09	*H. syriacus* ‘Strawberry Smoothie’	‘Lohengrin’	4*x*	4.76 ± 0.05	17.2
H2015-047-05	*H. syriacus* ‘Strawberry Smoothie’	‘Lohengrin’	4*x*	4.95 ± 0.13	25.4
H2015-052-02	*H. syriacus* ‘Blushing Bride’	‘Lohengrin’	4*x*	4.83 ± 0.05	13.7
H2015-052-05	*H. syriacus* ‘Blushing Bride’	‘Lohengrin’	4*x*	4.72 ± 0.02	4.8
H2015-052-12	*H. syriacus* ‘Blushing Bride’	‘Lohengrin’	4*x*	4.81 ± 0.06	29.4
H2015-052-X2	*H. syriacus* ‘Blushing Bride’	‘Lohengrin’	4*x*	4.79 ± 0.03	11.8
H2015-060-23	*H. syriacus* ‘White Chiffon’	‘Lohengrin’	4*x*	4.78 ± 0.03	40.1
H2015-061-03	*H. syriacus* ‘Lavender Chiffon’	‘Lohengrin’	4*x*	4.75 ± 0.01	31.9
H2015-062-08	*H. syriacus* ‘Lavender Chiffon’	‘Lohengrin’	4*x*	4.83 ± 0.04	27.8
H2015-064-10	*H. syriacus* ‘Strawberry Smoothie’	‘Lohengrin’	4*x*	4.76 ± 0.03	20.5
H2015-068-01	*H. syriacus* ‘White Chiffon’	‘Lohengrin’	4*x*	4.63 ± 0.09	24.6
H2015-072-10	*H. syriacus* ‘Lavender Chiffon’	‘Lohengrin’	4*x*	4.75 ± 0.04	35.6
H2015-076-08	*H. syriacus* ‘Blushing Bride’	‘Lohengrin’	4*x*	4.75 ± 0.04	23.7
H2015-081-01	*H. syriacus* ‘Blushing Bride’	‘Lohengrin’	4*x*	4.83 ± 0.02	27.6

^1^ A total of 18 progenies of ‘Resi’ and of 19 progenies of ‘Lohengrin’ were randomly chosen from a population that has been selected by general plant growth vigor. ^2^ Petal area measurement was based on two random chosen flowers [4]. * Individuals having a larger petal area than both of its parents.

**Table 2 genes-13-01022-t002:** FISH analysis results of rDNA loci for a *H. syriacus* cultivar standard, interspecific hybrid cultivars, ‘Lohengrin’ and ‘Resi’, and BC_1_F_1_ hybrids.

Cultivar Name or Accession	Female Parent	Male Parent	Predicted Ploidy Level	45S rDNA	5S rDNA
*H. syriacus* ‘Bali’ [13]	-	-	4*x*	4	2
‘Lohengrin’	-	-	4*x*	5	2
‘Resi’	-	-	4*x*	5	2
H2015-029-02	*H. syriacus* ‘Strawberry Smoothie’	‘Lohengrin’	4*x*	4	2
H2015-064-10	*H. syriacus* ‘Strawberry Smoothie’	‘Lohengrin’	4*x*	5	2
H2015-061-03	*H. syriacus* ‘Lavender Chiffon’	‘Lohengrin’	4*x*	5	2
H2015-052-X2	*H. syriacus* ‘Blushing Bride’	‘Lohengrin’	4*x*	6	2
H2015-016-08	*H. syriacus* ‘Blushing Bride’	‘Resi’	6*x*	5	3
H2015-016-03	*H. syriacus* ‘Blushing Bride’	‘Resi’	6*x*	6	3
H2015-017-02	*H. syriacus* ‘Blushing Bride’	‘Resi’	6*x*	6	3
H2015-017-05	*H. syriacus* ‘Blushing Bride’	‘Resi’	6*x*	6	3
H2015-017-08	*H. syriacus* ‘Blushing Bride’	‘Resi’	6*x*	6	3
H2015-019-09	*H. syriacus* ‘Blue Chiffon’	‘Resi’	6*x*	6	3
H2015-024-07	*H. syriacus* ‘Blue Chiffon’	‘Resi’	6*x*	6	3
H2015-109-01	*H. syriacus* ‘Lavender Chiffon’	‘Resi’	6*x*	6	3
H2015-016-01	*H. syriacus* ‘Blushing Bride’	‘Resi’	6*x*	7	3
H2015-024-01	*H. syriacus* ‘Blue Chiffon’	‘Resi’	6*x*	7	3
H2015-104-05	*H. syriacus* ‘Lavender Chiffon’	‘Resi’	6*x*	8	3

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
