# Peer review of "Near-Hexaploid and Near-Tetraploid Aneuploid Progenies Derived from Backcrossing Tetraploid Parents Hibiscus syriacus × (H. syriacus × H. paramutabilis)"

_genes, 2022, doi:10.3390/genes13061022_

Round 1

Reviewer 1 Report

The manuscript “Near-Hexaploid and Near-Tetraploid Aneuploid Progenies derived from backcrossing tetraploid parents Hibiscus syriacus × (H. syriacus × H. paramutabilis)” submitted by Chen and Contreras described a significant interspecific Hibiscus backcrossing breeding program. The authors have successfully tested a range of ploidy levels such as tetraploid, hexaploidy, and aneuploid for parental plants or their hybrids. The overall research protocol comprised a range of individual experiments including morphological analysis of traits, pollen staining tests and size measurement, hybridity authentication using molecular markers (ISSR, SRAP), DNA content measured by flow cytometry, chromosome count and FISH analysis. Subsequently, from such a systematic design, the authors not only examined the ploidy level of the parents and individual plants in the hybrid population, but also successfully revealed the cytogenetical chromosomal segregating pattern during hybrid formation. By using some basic experimental procedures, the authors have been able to evaluate their breeding targets, and measure the ploidy level of the BC1F1 population whilst locating the rDNA loci in the hybrid genome to confirm any chromosome rearrangement activity. Certainly, this paper would be a model for plant breeders especially those engaged in interspecific hybridization when the ploidy level becomes an issue. Finally, the English expression, phraseology and sentence structure in this paper makes it suitable after minor spell check.

Author Response

Thank you for your advice and support. I have had a native speaker in Plant breeding professional to do English editing after your advice. We appreciate your efforts for us to improve our article. 

Reviewer 2 Report

This study provided the breeding potentials of the interspecific hybrid Hibiscus cultivars for increasing flower size and ploidy level. My comments are:

  1. In my view, the novelty of this study is potential ploidy breeding. But in the introduction , the ploidy level of Hibiscus species, such as H. syriacus H. paramutabilis and H. syriacus was not described, the significance and strategy of ploidy manipulation in Hibiscus was not mentioned.
  2. Figure 1 and Figure should be listed in Result section not in Methods.
  3. For statistical analysis, the phenotypic data such as flower size should be measured at least three flowers of each plant, not two flowers.
  4. There are no supplement Table 1 and Table 2
  5. In the text, the author said ‘these plants have been through an early selection by their growth vigor and flower size. Thus, the observation of 78% progenies being hexaploid would not represent a random unselected population’(line 280-282),’ a larger scale test in a random population would be needed to confirm the relationship’ (line 306). Why the author did not test more progenies? In my opinion, all progenies having increased petal area should be detected ploidy level.
  6. The legend of Figure 3 is not clearly described.
  7. In Figures 4 and 5, what is the pink color indicated? What is arrow indicated?
  8. Discussion ‘Breeding potential of hybrid cultivars for large flower introgression’ is simple, authors should revise it thoroughly.
  9. The language should be polished.

Minor comments

  1. The name of cultivar should be in singlequotes, such as in line 16
  2. There are space between number and unit, such as line 218

The figure 1 in line 348 should be Figure 6.

Author Response

Thank you for the review. We appreciate your effort. We made editing and responses that are listed below (Marked in blue). 

  1. In my view, the novelty of this study is potential ploidy breeding. But in the introduction , the ploidy level of Hibiscus species, such as H. syriacus H. paramutabilis and H. syriacus was not described, the significance and strategy of ploidy manipulation in Hibiscus was not mentioned.

(1) Ploidy information of both species is added in line 59. Line 82-93 are all about the value of ploidy manipulation in Hibiscus, and lines 91-94 mention its value on H. syriacus

  1. Figure 1 and Figure should be listed in Resultsection not in Methods.

(2) Table 1 moved to results 3.5. Thanks 

  1. For statistical analysis, the phenotypic data such as flower size should be measured at least three flowers of each plant, not two flowers.

(3) Since the target of the statistics is to compare the flower size of each ploidy, not the flower size of each individual, the 3 repeat is not necessary. That is why we don't claim which genotype has a bigger flower; instead, we only mention the differences between ploidy. 

  1. There are no supplements Table 1 and Table 2

(4) We did upload supplement tables 1 and 2. Not sure if the system issue or something. I will let the editors know about this issue. I temperately add the supplement table in the end of the manuscript.

Thanks

  1. In the text, the author said ‘these plants have been through an early selection by their growth vigor and flower size. Thus, the observation of 78% progenies being hexaploid would not represent a random unselected population’(line 280-282),’ a larger scale test in a random population would be needed to confirm the relationship’ (line 306). Why the author didnot test more progenies? In my opinion, all progenies having increased petal area should be detected ploidylevel.

(5) It is unfortunate but we didn't consider assessing ploidy until after we selected the population and those unselected plants were discarded, meaning they are not available after selection. Because of this, we addressed the situation and honestly describe the ratio of hexaploids in the population that remains and should not be considered an estimate of the whole population. 

  1. The legend of Figure 3 is not clearly described.

(6) The description of Figure 3 is edited. Thanks 

  1. In Figures 4 and 5, what is the pink color indicated? What is arrow indicated?

(7) The red arrows aimed to point out the red signal (pink) and the green arrows were aimed to point out the green signal. We don't know why they are shifted but the issue has been corrected. I will also contact the editors to make sure this issue will not show again.

Thanks for pointing it out. 

  1. Discussion ‘Breeding potential of hybrid cultivars for large flower introgression’ is simple, authors should revise it thoroughly.

(8) The breeding potential of introgression in Hibiscus has been discussed in the introduction part of Lines 59-70. We also re-editing the whole paragraph of the ‘Breeding potential of hybrid cultivars for large flower introgression’.

  1. The language should be polished.

(9) Language has been polished again before the re-submission. Thank you

Thanks again for helping us to improve this article. We appreciate sincerely it.  

Round 2

Reviewer 2 Report

The manuscript was finely revised. I agree to accept.

Author Response

Thank you for your earlier suggestions. We did another run of editing on our language and format. We sincerely appreciate your effort. 

Hsuan